# Psychotherapeutic Techniques for Distressing Memories: A Comparative Study between EMDR, Brainspotting, and Body Scan Meditation

**DOI:** 10.3390/ijerph19031142

**Published:** 2022-01-20

**Authors:** Fabio D’Antoni, Alessio Matiz, Franco Fabbro, Cristiano Crescentini

**Affiliations:** 1Maternal Infant Services Unit of Udine, Azienda Sanitaria Universitaria Integrata Friuli Centrale (ASUFC), 33100 Udine, Italy; 2Department of Languages and Literatures, Communication, Education and Society, University of Udine, 33100 Udine, Italy; alessio.matiz@uniud.it (A.M.); franco.fabbro@uniud.it (F.F.); cristiano.crescentini@uniud.it (C.C.); 3Department of Psychology, Sapienza University of Rome, 00118 Rome, Italy; 4Institute of Mechanical Intelligence, Scuola Superiore Sant’Anna di Pisa, 56010 Pisa, Italy

**Keywords:** psychotherapy, distressing memories, EMDR, Brainspotting, body scan meditation, mindfulness, bottom-up therapy, body-oriented intervention, trauma, stress

## Abstract

Objectives: We explored the effects of a single 40-min session of Eye Movement Desensitization and Reprocessing (EMDR), Brainspotting (BSP), and Body Scan Meditation (BSM) in the processing of distressing memories reported by a non-clinical sample of adult participants. Design: A within-subject design was used. Methods: Participants (*n* = 40 Psychologists/MDs) reported four distressing memories, each of which treated with a single intervention. EMDR, BSP, and BSM were compared with each other, and with a Book Reading (BR) active control condition, using as dependent measures, SUD (Subjective Units of Disturbance) and Memory Telling Duration (MTD) on a 4-point timeline: Baseline, Pre-Intervention, Post-Intervention, Follow-up. Results: SUD scores associated with EMDR, BSP, and BSM significantly decreased from Pre- to Post-Intervention (*p* < 0.001). At Post-Intervention and Follow-up, EMDR and BSP SUD scores were significantly lower than BSM and BR scores (*p* < 0.02). At both Post-Intervention and Follow-up, BSM SUD scores were lower than BR scores (*p* < 0.01). A reduction in MTD was observed from Pre- to Post-Intervention for EMDR and BSP conditions (*p* < 0.001). Conclusions: Overall, results showed beneficial effects of single sessions of EMDR, BSP, or BSM in the processing of healthy adults’ distressing memories. Study limitations and suggestions for future research are discussed.

## 1. Introduction

In common experience, people live with disturbing, yet not necessarily traumatic, memories. Remarkably, the more people try to push away these memories, the more they tend to come back and contribute to causing discomfort. Fortunately, experiences like this, however painful, are common among people, and are not necessarily signs of psychopathology.

In clinical contexts, however, distressing or disturbing memories can consist of images, thoughts, and feelings related to a traumatic or stressful event. These memories are generally negatively emotionally charged, and may originate from both “big trauma” (“T”), such as life-threatening experiences and sexual violence, or “little trauma” (“t”) or relational trauma, which include adverse, extremely upsetting life events, such as non-life-threatening injuries, childhood humiliations, death of a pet, bullying, and loss of significant relationships [1]. Distressing memories related to “T” or “t” may be triggered or occur spontaneously, and are a distinguishing characteristic of Post-Traumatic Stress Disorder [2] (PTSD). In the context of trauma-focused interventions, there are several therapeutic approaches recommended to address directly distressing memories, such as Prolonged Exposure and trauma-focused Cognitive Behavioral Therapy [3].

However, in recent decades, other techniques that can be integrated in a psychotherapeutic intervention have emerged that try to help patients to lessen emotionally upsetting activation during the recalling of distressing or traumatic experiences associated with present symptoms. One of the most studied techniques in this context is Eye Movement Desensitization and Reprocessing [4], which is now considered as the elective psychotherapy for the treatment of PTSD in children, adolescents, and adults [5]. The theoretical framework of EMDR is based on the Adaptive Information Processing (AIP) model [1]. According to this model, the memory of a distressing event is stored in an unprocessed and maladaptive form, as if it were frozen, and thus, unable to connect with other memory networks that hold adaptive information. The assumption is that EMDR therapy allows the accessing of the distressing or traumatic memory network, and facilitates the processing of the event, bringing it to an adaptive resolution.

The EMDR procedure involves eight phases: (1) history taking, (2) preparation, (3) assessment, (4) desensitization, (5) installation, (6) body scan, (7) closure, and (8) reassessment [1]. During phase 3, the distressing memory processed is assessed by identifying its worst connected mental image and its current manifestations at cognitive, affective, and somatic levels. During desensitization and processing phases (4–6), the distressful memory is processed using dual-attention Bi-Lateral Stimulation (BLS), most commonly alternate horizontal Eye Movements (EMs). 

A series of studies have indicated that EMs during memory retrieval reduce memory vividness and emotional activation [6,7,8,9]. This reduced vividness was assessed not only through self-report measures, but also via behavioral tasks [10] (i.e., Reaction Time task), and with functional magnetic resonance imaging [11]. Several theories have been proposed to explain possible EMDR underlying mechanisms of action, such as the orienting and relaxation response hypothesis [6,12], the limited working memory resources theory [13], and more complex neurobiological models [14,15,16,17]. Although an ongoing controversy is present about the possible critical role of eye movements in the processing of distressing memories, clinical practice and neurobiological findings [18,19] suggest that EMs are likely only a part of a larger and more complex clinical intervention for the treatment of disturbing memories.

Another well-known therapeutic technique, but less supported by research than EMDR, is Brainspotting (BSP). It was developed in 2003 by David Grand, as a result of his EMDR work (see “Natural Flow EMDR”) [20] and training in “Somatic Experiencing” [21,22]. In BSP, it is suggested that the direction in which people look or gaze can affect the way they feel [23]. 

During a typical BSP session, the therapist guides, through a pointer, the eyes of the client across the field of vision to find an appropriate eye position (“Brainspot”) to “activate” the psychophysiological response to a traumatic memory. As opposed to EMDR, where the traumatic memory is the “target” of treatment, in BSP, the target is the visual point of activation. The purpose is thus to identify the Brainspot as this visual point that appears to promote the client’s processing of the distressing or traumatic experience, and of, as in the EMDR approach, the memories, thoughts, or sensations connected to it. Unlike processing with EMDR, in BSP, the distressing memory is processed without following specific series of steps or verbal reports. For example, whereas in EMDR, the client generally receives intermittent BLS and has the possibility to share what she/he has been noticing, in BSP, the client continuously listens to a slowly acoustic BLS (“BioLateral Sound”) during the processing, and does not necessarily have to talk to the therapist during elaboration. Moreover, compared to EMDR, the processing phase in BSP is generally more focused on body sensations. Indeed, in many cases, the therapeutic intervention during the memory processing is limited just to bringing the client’s attention back to his/her own body.

Nonetheless, a common feature of EMDR and BSP is that both invite clients to mindfully pay attention to their inner experience. For example, during EMDR, clients are invited to just notice what is occurring in the present moment, and to let whatever happens, happen [24]; similarly, in BSP, the importance of the attuned, mindful, and compassionate presence of the BSP therapist is emphasized, as well as the client’s mindful state during the processing [23]. Generally speaking, these characteristics appear to be consistent with contemporary conceptualizations of the mindfulness construct and the relative mindfulness practice in both the clinical/psychotherapeutic and non-clinical contexts [25,26].

More generally, in the field of mindfulness and mindfulness meditation, Body Scan Meditation (BSM) is a body-centered practice transversal to several popular mindfulness-based interventions aimed at reducing individuals’ stress, and relieving their suffering. BSM is usually considered a focused-attention meditation practice [27]. This kind of practice involves voluntarily shifting one’s attention first to specific body parts (e.g., toes, back, or head), and then to the whole body, in order to notice what is happening (e.g., sensations such as pain or muscle tension) in the present moment without judging or reacting to the experience (equanimous attitude) (e.g., MBSR) [28]. BSM is significantly related to the mindfulness facets of observing and non-reacting to inner experience, as well as to improvements in psychological well-being [29], and thus, appears to be an important clinical resource that one could integrate into psychotherapeutic work [30]. Nevertheless, BSM is not typically used in association with disturbing memories (differently from EMDR and BSP), and, as such, does not specifically require individuals to follow their stream of consciousness related with these memories, but only to notice when the mind wanders, in order to bring it back to body sensations. 

Overall, despite the increasing amount of research in clinical and non-clinical samples, the current understanding of the mechanisms of action of EMDR is still limited [31]. Even more, although BSP is increasing in popularity among therapists, and some interesting neurobiological hypotheses on its underlying mechanisms have been recently proposed [32,33], there is still a paucity of literature evaluating the effectiveness of this therapeutic tool [34,35]. Moreover, the attempts to directly compare BSP with EMDR are very limited [36]. 

Therefore, the aim of the current study was to explore and compare the effects of a single 40-min session of EMDR, BSP, and BSM techniques in the processing of distressing memories shown by a non-clinical sample of adult participants (medical doctors or psychologists in training as psychotherapists). A non-clinical sample was chosen because we intended to reduce any potential negative effect of eventual severe psychopathology, neuropsychological signs, or psychopharmacological treatment, on both the therapeutic relationship and the capacity of participants in self-focusing on their inner experience and recollecting memories; moreover, we also intended to limit the possible adverse effects potentially originating from the (brief) treatment of participants’ distressing memories. 

Since clinical observations suggest that EMDR standard protocol (which generally incorporates the use of EMs) stimulates accelerated memory reprocessing in just 40 min of treatment [1]; the primary hypothesis of the study was that EMDR would have been more effective in reducing memory-associated distress than BSP or BSM techniques. Indeed, the application of BSP in clinical settings generally requires longer treatments that exceed one h; moreover, as previously mentioned, BSM is not specifically designed for the desensitization and reprocessing of painful memories. In comparing the three techniques in a non-clinical sample, we aimed to provide some initial and preliminary evidence about what could be crucial elements of effective interventions in the processing of distressing memories. In this respect, a possibility could be linked to mindful observation of the stream of consciousness and somatic sensations linked to the processed memories. To evaluate this hypothesis, we also introduced an active control condition, in which participants were engaged in reading a book about trauma, that is in an activity not designed to target the core therapeutic features of interest.

## 2. Materials and Methods

### 2.1. Participants

Participants were psychologists and medical doctors (NP = 37, NMD = 3) attending a four-year specialization in Systemic Psychotherapy at an Italian Institute of Family Therapy (Females *n* = 34, Males *n* = 6; Age: M = 34.61, SD = 7.13 years). We approached 48 persons, 42 of whom initially agreed to participate in the research project. Before the actual start of the project, a participant withdrew from the research, and the last participant agreed to be not included in the study due to the completion of the Williams experimental design reached at the fortieth participant. Participants had theoretical, but not practical, knowledge of EMDR, BSP, and BSM. The criteria for their inclusion in the study were: not suffering of a severe clinical syndrome or a personality disorder (self-reported by participants, and assessed in the screening phase of our study; see below), and having a normal neuropsychological profile in the areas of memory and lexical access (also assessed in the screening phase). For possible adverse effects resulting from the processing of the distressing memories, we tested the interventions on a sample of Psychologists and MDs willing to seek help from a therapist if necessary. The professional experience of the participants also made it possible to tell the researcher, in the follow-up session, their personal impressions about the interventions and the overall experiment. Informed consent for the research assessment and procedure was obtained from all participants. The procedures were approved by the local Ethics of the University of Udine, and were in accordance with the Helsinki Declaration guidelines.

### 2.2. Screening Phase Measures

Millon Clinical Multiaxial Inventory (MCMI-III): The Italian version of the MCMI-III is a self-report personality questionnaire composed of 175 true–false items (for example, item 40: “I guess I’m a fearful and inhibited person.”) that shows Cronbach’s alpha coefficients mainly comparable to those in the original American version (i.e., α = 0.73–0.95) [37,38]. Based on the DSM-IV classification system, MCMI-III assesses personality patterns and clinical syndromes. In our study, the Base Rate scores in the Severe Clinical Syndromes scales (Thought Disorder, Major Depression, Delusional Disorder) and in the Severe Personality Disorders scales (Schizotypal, Borderline, and Paranoid) were computed.

Italian Short Neuropsychological Assessment (Esame Neuropsicologico Breve 2-ENB2): The ENB2 is a battery of fifteen screening neuropsychological tests for adults [39]. Only two tasks were used: (1) the short story test [40] to evaluate long-term verbal memory (participants were told a story consisting of 28 basic elements, and asked to tell it again immediately after listening, and after 10 min; the test score consisted of the number of elements recalled during the participants’ second retelling), and (2) the phonemic fluency test to assess individuals’ lexical access and retrieval (participants were asked to tell words beginning with the letters C, P, and S; for each of the three letters, one min was given to name as many words as possible; the test score was the average number of words generated in the three lists). 

Rey–Osterrieth Complex Figure Test (ROCF): The ROCF test [41] was employed to assess long-term visual memory. Participants were requested to observe and copy a complex line drawing. Scoring was obtained on the basis of the delayed reproduction of the figure after 10 min. Individual raw scores were converted to Equivalent Scores (ranging from 0 to 4) on the basis of the age-, education-, and gender-matched average scores of a large control sample of 280 healthy adult individuals [42].

### 2.3. Disturbing Memories Measures

Subjective Units of Disturbance (SUD) Scale: The SUD Scale is a one-item subjective distress scale ranging from 0 to 10, originally developed by Wolpe [43]. In our study, the SUD Scale was used to measure subjective distress experienced after each memory telling. Participants were asked to rate on a 0-to-10 scale how disturbed they were by the memory they had just told the researcher (i.e., “On a scale of 0–10, where 0 is not disturbance/distress or neutral and 10 is the highest disturbance one can imagine, how disturbing does the memory feel to you now?”). The SUDS were completed the same number of times in all of the conditions. During memory processing with EMDR and BSP, we generally used the term “activation” instead of “distress/disturbance” (see below for more details on EMDR and BSP interventions). Psychotherapy outcome research supports SUD scores as a global measure of distress level [44] with good psychometric properties [45]. In line with the EMDR approach [1], SUD scores are considered “high” when rated ≥6.

Memory Telling Duration (MTD): In all experimental sessions (described below), we measured the length of time a participant spent recounting each memory to the researcher. This measure was hypothesized to be connected to the SUDS assuming that, as the memory processing took place, it was possible to find both a decrease of SUD scores, and a reduced length of time spent to recounting the memory.

### 2.4. Procedure

The research was carried out through the following phases (see Table 1): Screening, Baseline (with the first verbalization of memories), Interventions (with verbalization of memories at Pre- and Post-Intervention), and Follow-up (with the final verbalization of memories).

First, all participants underwent three screening tests (MCMI-III, ENB2, and ROCF) to assess eligibility. The MCMI-III criteria of inclusion were: not having a Base Rate score equal to or greater than the clinical cut-off (BR ≥ 85) in the severe Clinical Syndromes scales (“Thought Disorder”, “Major Depression”, “Delusional Disorder”), or in the severe Personality Disorders scales (“Schizotypal”, “Borderline”, and “Paranoid”). Each participant was also required to achieve normal scores in ENB2 evaluation (i.e., above the 5th percentile of the reference population) [39]. Regarding the ROCF, the exclusion criterion was an Equivalent Score lower than 1.

At baseline, each participant was asked to tell the researcher four distressing memories. Participants could choose both recent and ancient memories belonging to their personal history (e.g., family wounds, workplace stress, bereavement, accident, hospitalization, maltreatment, or witnessed violence, etc.). The instructions given to the participants were to freely tell the memory to the experimenter with no time limits. At the end of each memory, the participant reported the SUD score, waited a few minutes to manage the emotional activation, and then moved onto the next memory. 

During the intervention phase, each participant received four experimental interventions: two therapeutic techniques (EMDR and BSP), a mindfulness meditation (BSM), and a control intervention (Book Reading, BR). Each intervention lasted about 40 min, and targeted one of the four memories told by the participant during the Baseline phase. The within-participant association between memory and intervention was pseudo-random in order to balance the memories’ baseline SUD scores across the four interventions. About one week passed from the initial screening phase to the Baseline session, and another week from the Baseline session to the start of the Intervention phase. Each participant received one intervention per week, following the sequence of a 4 × 4 Williams design (see Table 2).

On the same meeting of the intervention (i.e., at the Pre-Intervention session; see Table 1), the participant was informed which of his/her memories told during the Baseline phase would have been processed on that day; s/he was then asked to retell that memory (as if it were the first-time s/he did it) and, immediately afterwards, to report the current SUD score for that memory. Immediately after receiving the intervention (at the Post-Intervention session), the participant was asked to tell the same memory again, and to report the current SUD score for that memory.

Approximately two months after receiving the last intervention, each participant met the researcher again, and was asked to retell his/her four memories, in the same sequence as during the Baseline phase, and to report the current SUD scores after each memory telling (the Follow-up session; see Table 1).

More specifically, the intervention protocol consisted of the following four types of interventions (see Table 3 for details).

EMDR: The EMDR procedure involved the following phases for treating the distressing memory. In accordance with phase 3 of the EMDR Standard Protocol [1], the therapist asked the participant to: (i) describe the worst mental image of the event; (ii) identify a current negative cognition about the self, linked to the memory; (iii) choose an alternative self-referred positive cognition, checking its validity (how much s/he felt the positive cognition to be true on a scale from 0 to 7); (iv) describe what emotions are brought up in connection with the recall of the distressing memory; (v) assess the intensity of the distress (trough the SUD scale mentioned above); and (vi) report the body locations of any perceived disturbance. Next, in line with phases 4–6 of the EMDR Standard Protocol, the therapist proceeded to the desensitization and reprocessing of the distressing memory using BLS. For BLS, we primarily used Ems; tapping was used only whenever necessary (i.e., during crying jags). In case of complete memory desensitization (i.e., the participant reported an activation of 0 on a scale of 0 to 10 while focusing on the initial memory), the therapist proceeded to the installation of the identified positive self-referred cognition about the memory, and checked for any residual disturbance perceived by the participant in his/her body. On average, EMDR sessions lasted 41.45 min (Table 3).

BSP: The “Inside Window” Brainspotting technique was used [46]. The therapist held a pointer of about 80 cm, and asked the participant to fix the end. The therapist asked the participant to feel in his/her body the highest activation linked to the distressing memory; the therapist then slowly moved the pointer and guided the participant’s eyes through his/her visual field, asking him/her to continue listening for their body’s activation. When the eye position associated with the maximum activation in the body was found (Brainspot), the therapist stopped moving the pointer, and asked the participant to fix the gaze there. The therapist then proceeded to the processing phase, supporting the participant’s self-observation (“focused mindfulness”) within a “dual attunement” frame [46]. In Brainspotting therapy, the expression “dual attunement” refers to a process supposed to be both relational and neurological, through which the therapist continuously tries to remain connected to the therapeutic relationship, as well as to the client brain-body response in therapy. According to this approach, the attuned, empathic, witnessing presence of the therapist promotes adaptive changes in the client [23]. During BSP, the participant continuously received a slow bilateral acoustic stimulation (a recording of ocean waves was played at a background volume) via headphones. In case of complete memory reprocessing (i.e., the participant reported an activation of 0 on a scale of 0 to 10 while focusing on the initial memory), the therapist asked the participant to bring attention back to the initial memory to assess any changes, and then proceeded to the “Squeeze the Lemon” technique (i.e., “Go inside and try to push the distress level as high as you can”) [46] until the participant felt no residual emotional activation. On average, BSP sessions lasted 40.03 min (Table 3).

BSM: We used the MBSR version of BSM [28]. The therapist asked the participant to simply sit on a chair, and notice what it feels like to be connected to the ground. Then, the therapist proceeded to guide BSM. In this practice, the individual focused on the feelings and sensations of specific parts of the body, systematically moving attention from one part to another until paying attention to the body in its totality at the end of the session. During the focusing on a specific body part, the therapist invited the participant to mentally breathe in that region. The participant was also asked to bring attention back to the body whenever s/he realized that her/his mind was distracted by something else (e.g., thoughts, emotions, memories, etc.). Each BSM lasted for a fixed time of 40 min.

BR: As a control condition, we chose a distracting cognitive activity, such as reading a book about traumatic stress (van der Kolk, B.A. (2014). The body keeps the score: Brain, mind, and body in the healing of trauma. Viking). The psychotherapist was in the room with the subject throughout the 40-min reading session.

Overall, although the duration of the BSM and BR sessions was fixed, the slightly variable durations of the EMDR and BSP sessions depended on the individual processing times. These times were agreed upon by the therapist and the participant based on the progress of the memory processing. In regard to the psychotherapist who delivered all the interventions and the control condition, he was an EMDR practitioner, trained in BSP as well, and an MBSR teacher with over 10 years of clinical practice (FD, the first author of this article). The therapist was thus trained at an advanced level in each of the three techniques used in the study, which, moreover, are commonly used by him in his ordinary clinical activity.

### 2.5. Statistical Analyses

After memory telling at Baseline, each memory was pseudo-randomly assigned to one of the four interventions (EMDR, BSP, BSM, BR). In two 4 × 4 ANOVAs with repeated measures, one for SUDs and the other for MTD values, the within-subject factors were Session (“Baseline”, “Pre-Intervention”, “Post-Intervention”, “Follow-up”) and Intervention (“BR”, “BSM”, “BSP”, “EMDR”).

The Greenhouse–Geisser estimate of sphericity was used to correct for violations of homogeneity of variance (indicated as *p*[GG]). Adjustments for multiple comparisons of post-hoc tests were performed using Bonferroni/Holm correction. The significance threshold of *p* = 0.05 was adopted in all analyses. All analyses were performed using the free software environment, R. The data that support the findings of this study are available from the corresponding author, upon request.

## 3. Results

### 3.1. Baseline Profile of Participants

The baseline profile of participants is detailed in Table 4. All participants scored below the clinical cut-off level in the severe Clinical Syndromes and Personality Disorders scales of MCMI-III. In ENB2, all participants scored above tolerance limits (the 5th percentile) for long-term verbal memory levels and for phonemic fluency. In the assessment of visuospatial memory using ROCF, all participants scored between the values of 1 and 4 (Equivalent Score). In sum, baseline profiles of participants showed that they had no manifest psychological or personality disorders, and intact long-term memory and lexical access functions.

### 3.2. Subjective Units of Disturbance

At baseline, mean SUD values for the memories treated with the three types of interventions (EMDR, BSP, BSM) and with the control condition (BR) were higher or equal than 6, which is the typical value considered to reflect a memory with a high negative emotional charge in the EMDR approach [1] (see Table 5). In particular, at baseline, no memory reported by participants was associated with a SUD less than 6 in the present study.

The 4 × 4 ANOVA highlighted a main effect of Session (F (3, 117) = 21.0, *p*[GG] < 0.001), a main effect of Intervention (F (3, 117) = 176.8, *p*[GG] < 0.001), and an interaction effect between Session and Intervention (F (9, 351) = 23.3, *p*[GG] < 0.001) (see Figure 1 and Table 5). Post-hoc tests for the main effect of Session revealed that SUD scores at Baseline were not different from scores at Pre-Intervention (*p* = 1.00), but that these scores significantly decreased from Pre- to Post-Intervention (SUD_Pre > SUD_Post, *p* < 0.001), and also from Post-Intervention to Follow-up (SUD_Post > SUD_Follow-up, *p* = 0.045). Post-hoc tests for the main effect of Intervention indicated that SUD scores associated with the EMDR, BSP, and BSM techniques were significantly lower than SUD scores associated with BR (SUD_EMDR < SUD_BR: *p* < 0.001; SUD_BSP < SUD_BR: *p* < 0.001; SUD_BSM < SUD_BR: *p* = 0.042).

Post-hoc tests for the interaction effect between Session and Intervention showed that: the Baseline vs. Pre-Intervention comparison of SUD scores was not significant for all Interventions (*p* = 1.00); the Pre- vs. Post-Intervention comparison was significant for the EMDR, BSP, and BSM techniques (SUD_Pre > SUD_Post, *p* < 0.001), but not for BR (*p* = 0.15); the Post-Intervention vs. Follow-up comparison was significant for BR (SUD_ Post > SUD_Follow-up, *p* = 0.001), but not for the EMDR, BSP, and BSM techniques (*p* = 1.00). Post-hoc tests for the interaction effect between Session and Intervention also revealed that: at Baseline and at Pre-Intervention, all between-intervention comparisons of SUD scores were not significant (*p* > 0.73); at Post-Intervention, SUD scores associated with BR were significantly higher than those associated with BSM, BSP, or EMDR (*p* < 0.001); SUD scores associated with BSM were significantly higher than those associated with BSP or EMDR (*p* < 0.02), whereas SUD scores associated with BSP were not different than those associated with EMDR (*p* = 0.41); at Follow-up, the multiple comparisons revealed the same pattern of results observed at Post-Intervention.

### 3.3. Memory Telling Duration

The 4 × 4 ANOVA highlighted a main effect of Session (F (3, 117) = 14.2, *p*[GG] < 0.001), and an interaction effect between Session and Intervention (F(9, 351) = 2.6, *p*[GG] = 0.04). Post-hoc tests for the main effect of Session revealed that MTD values at Baseline were not different from values at Pre-Intervention (*p* = 1.00), and that the MTD values significantly decreased from Pre- to Post-Intervention (MTD_Pre > MTD_Post, *p* = 0.001), but not from Post-Intervention to Follow-up (*p* = 1.00) (see Table 5). Post-hoc tests for the interaction effect between Session and Intervention showed that the Baseline vs. Pre-Intervention comparison of MTD values was not significant for each Intervention (*p* > 0.15); the Pre- vs. Post-Intervention comparison was significant for the EMDR and BSP techniques (MTD_Pre > MTD_Post, *p* < 0.001), but not for the BSM technique nor for BR (*p* > 0.33); and the Post-Intervention vs. Follow-up comparison was never significant for any Intervention (*p* > 0.27). Post-hoc tests for the interaction effect between Session and Intervention also revealed that, at each Session, no between-intervention difference was observed (for all, *p* > 0.14).

## 4. Discussion

The purpose of this study was to explore the effects of 40-min single-session interventions with EMDR, BSP, and BSM techniques on the individuals’ ability to process negative life events. In a within-subject design, participants told the researcher four distressing memories, each of which was treated with a single intervention (EMDR, BSP, BSM, and a book reading control condition, BR). All interventions were delivered in a non-clinical sample of psychologists and MDs. These professionals were initially screened for psychopathology and neuropsychological functioning with neuropsychological tests and self-report questionnaires. We compared the three interventions and the active control condition using both Subjective Units of Disturbance (SUD) and Memory Telling Duration (MTD) measures on a 4-point timeline (sessions): Baseline (i.e., about one week before the intervention), Pre- and Post-Intervention (i.e., immediately before and after the specific intervention or control condition), and Follow-up (i.e., about two months after the intervention).

The analysis of SUD scores revealed a main effect of Session: SUD scores were not different between Baseline and Pre-Intervention, but they decreased from Pre- to Post-Intervention, and from Post-Intervention to Follow-up. More specifically, an interaction effect between Session and Intervention revealed that SUD scores associated with EMDR, BSP, and BSM, but not BR, techniques significantly decreased from Pre- to Post-Intervention. This indicates that all the three techniques were more effective than the control condition in mitigating the distress linked to the distressing memories. Therefore, social sharing of painful memories (arguably the most important “ingredient” in the BR condition) was not enough to significantly reduce subjective distress linked with the reported memories. Nonetheless, the exploration of the interaction effect revealed that SUD scores associated with BR decreased from Post-Intervention to Follow-up, although, at the final session, they remained significantly higher than the scores associated with the EMDR, BSP, and BSM interventions.

More specifically, regarding the primary hypothesis of the study on the greater effectiveness of EMDR in treating distressing memories, the obtained results appear to only partially confirm our expectation: indeed, immediately after the interventions, SUD scores associated with EMDR, but also with BSP, were significantly lower than scores associated with BSM and BR; the same pattern was observable at Follow-up. Although scores associated with EMDR at Post-Intervention and Follow-up were lower than the scores associated with BSP, this difference was not statistically significant. As far as the specific experimental design employed in the current study is concerned, EMDR and BSP thus appeared to be comparable in terms of efficacy in reducing healthy participants’ subjective disturbance connected with distressing memories. Trying to generalize these preliminary findings to the clinical setting, it could be important to take the client’s treatment preferences into particular attention during the therapeutic decision-making process. Indeed, an informal debriefing procedure conducted at the end of the present study, i.e., during the follow-up session, showed that some participants preferred BSP over EMDR because they had felt freer during memory processing in the former than the latter type of intervention. Nonetheless, other participants reported that they had felt “a little bit lost” during BSP, and preferred to be guided more by the therapist during memory processing using EMDR.

Though not specifically designed for a psychotherapeutic purpose, BSM was also effective, although to a lesser extent than EMDR and BSP, in reducing SUD scores linked to distressing memories at Post-Intervention, and this effect remained substantially stable in time (at Follow-up). In general, it is important to keep in mind that our findings cannot be directly extended to the whole therapeutic process from which the EMDR, BSP, and also BSM techniques can be extrapolated.

A possible explanation of the effectiveness of EMDR, BSP, and, to a lesser extent, BSM, in reducing SUD scores connected to distressing memories of healthy individuals could be attributable to two core components that characterize all these three interventions: (1) the degree of connection in the participant/therapist relationship, and (2) the focus on body sensations. Indeed, although in the first two conditions (EMDR and BSP), the relational involvement of the therapist is recognized as a key element of the intervention, in BSM, connection and intimacy may have been favored, at least to some extent, by the soothing prosody of the therapist’s guiding voice. Although this study employed the same therapist across all types of intervention to minimize the effects of potentially confounding variables related, for instance, to different therapists’ relational skills and personality characteristics, future studies interested in comparing different therapeutic techniques could attempt to include measures of therapeutic relationship for each of them, such as the degree of connection and attunement (e.g., the Working Alliance Inventory) [47].

Regarding focusing on body sensations, body self-observation with a non-judgmental, equanimous attitude is a transversal element both in the EMDR and BSP therapeutic techniques, and in BSM. According to the sensorimotor model of memory [48,49], telling a distressing memory could reactivate the somatosensory components of perceptual and motor information registered during the encoding of the event, and leave traces of activation in the body. In the specific case of the BSM, mindful observation of bodily sensations could affect these somatic traces connected to the memory, favoring an implicit processing of the experience, even without voluntarily thinking about it. Indeed, in the current study, several participants not only reported feelings of relief immediately after BSM, but also sometimes referred insights and different points of view on the distressing event during the Post-Intervention memory retelling. These individuals’ feedbacks thus indicate the therapeutic potential of focusing on one’s body parts and sensations when different physiological states are activated by the recalling of a distressing memory. More generally, these data suggest a deepening of the clinical implications of BSM [30]. For example, if a client shares a highly distressing memory during a therapeutic session, the therapist could guide a BSM practice before proceeding with the deepening of the experience and its elaboration with more specific techniques, such as EMDR. In this case, it could be possible that BSM could favor, to some extent, the spontaneous desensitization and processing of the event, although it does not require focusing on the specific memory or talking about it.

As mentioned above, body self-observation is a fundamental element not only of BSM, but also of EMDR and BSP. The basic element of mindful focusing on body sensations can also be found in other well-known bottom-up therapeutic approaches, such as Somatic Experiencing [21] and Sensorimotor Psychotherapy [50]. However, only in EMDR and BSP approaches is BLS is added during the processing, and, furthermore, self-observation is extended to the entire stream of consciousness connected to the target event (e.g., cognitions, images, feelings, or Brainspot). It could possible that eye movements in EMDR and visual fixation point in BSP Brainspot affect sensorimotor patterns activated during memory processing. Indeed, there is a large amount of literature on how eye movements and gaze affect memory retrieval [6,9,49,51].

Another possible explanation of the effectiveness of, in particular, EMDR and BSP in reducing SUD scores connected to distressing memories may be connected to imaginative exposure. Indeed, although not to the same extent as with other techniques such as exposure therapy, these two techniques imply a kind of imaginative exposure to the distressing memories, and this can contribute to their adaptive re-processing [52]. Remarkably, such a possibility could also explain, at least in part, the reduction of SUD scores in the control BR condition. Indeed, reading a book about trauma may have triggered the memory of one’s own distressing event, thus producing a limited imaginative exposure to it.

In summary, mindful self-observation of one’s stream of consciousness (especially body sensations, but also cognitions and images) connected to the distressing memory, in association with BLS in a context of a deep connection between client and therapist, seems to be a crucial element for developing effective interventions to mitigate and process psychological suffering linked with distressing memories.

Concerning MTD, data analysis showed that time of narration decreased overall from Pre- to Post-Intervention, but not from Baseline to Pre-Intervention, nor from Post-Intervention to Follow-up. The analysis of the interaction effect between Intervention and Session revealed that the reduction in MTD observed from Pre- to Post-Intervention was specifically due to the EMDR and BSP conditions. This indicates that after the application of these two therapeutic techniques, participants tended to tell their memories in a more concise way. The significant reduction in MTD occurred only at the Post-Intervention session, where the SUD scores associated with EMDR and BSP were lower than those in the BSM and BR conditions. It could be assumed that the therapeutic processing promoted by EMDR and BSP directly affected how the memories were reported. On the one hand, EMDR and BSP could require a greater involvement of the participants with respect to BSM and BR, and this could make it more tiring for individuals to retell the memory immediately after the intervention. On the other hand, the reduced length could also be an indicator of a more integrated and processed memory narrative that does not require further turns of words in order to be explained. The length of a narrative is an indicator already used in psychological research. Indeed, according to Grice’s “maxim of quantity” [53], and the identified relations between language and an individual’s attachment style [54], an overly verbose and not necessarily informative narrative may be an indicator of an adult “Preoccupied State of Mind”. In the context of our research, a micro- and macro-linguistic analysis of the transcribed memories is ongoing to explore possible links between memory processing and quality of the narrative [55].

The present study has a number of limitations. First of all, the small sample size and the specificity of the study population should be considered. Indeed, participants were psychologists and medical doctors attending a specialization in psychotherapy, and their beliefs and expectations with respect to the effectiveness of the various techniques used could have affected results (e.g., amplifying the effectiveness of experimental techniques compared with the control condition). However, participants’ possible positive anticipations on the experimental techniques might only partially explain the outcomes of the present study, since EMDR and BSP were found to be more effective than not only BR, but also BSM. Nonetheless, to overcome this limitation concerning the selected sample, it would be useful that future studies attempt to extend the present findings to larger groups of “non-professionals” participants, who, with caution, could be recruited outside psychotherapeutic trainings. Another limitation of our study may be related to the choice of the book used in the control condition. Indeed, reading a book on trauma may have led participants to implicitly equate their distressing, but not traumatic, memories to more serious traumas, thus contributing to evaluating their memories as still disturbing at the Post-Intervention phase. Therefore, it would be useful to carry out further studies with other neutral control conditions. In the present study, although a female participant reported side effects during EMDR intervention (i.e., fear of losing control of her mind), which quickly went into remission after an EMDR session with her referral therapist, in general, all participants reported feeling better after the interventions, and felt that they did not need further psychotherapy sessions focused on the memory being treated in the context of the study. More importantly, in the Follow-up session participants generally reported that they noticed positive effects in daily life after the intervention phase, especially in the days following EMDR and BSP sessions (e.g., feeling less anxious or less frustrated in relationships, etc.). Several participants also reported to have better understood some of their usual functioning patterns, and to have identified core psychological themes to be explored in a future path of individual psychotherapy. Despite these positive effects, it remains to be investigated to what extent the present data obtained after single sessions of processing of stressful memories are informative about the more complex and long therapeutic paths generally conducted in the clinical setting with, in particular, EMDR and BSP. Thus, this study underlines the importance of conducting further research comparing the effects of different therapeutic interventions on the processing of distressing/traumatic memories in clinical populations. For example, it would be interesting to study which specific features of therapeutic interventions could be most useful for each specific personality disorder. From this point of view, some questions may arise: could EMDR, as a structured protocol, potentially be more useful than BSP in borderline personality patterns? Or might it be more useful to use the more “open”, less structured approach of BSP with avoidant clients? Or are there times, depending on the development of the therapeutic relationship, for which it would be desirable to use one kind of intervention first, and then another one? Future research may help answer questions like these.

## 5. Conclusions

Our research produced relevant results showing the beneficial effects of single 40-min sessions of EMDR, BSP, or BSM on non-clinical individuals’ processing of distressing memories. We found that EMDR and BSP were comparable in terms of efficacy, and similarly affected how the distressing memories were reported in terms of conciseness. These results were primarily attributed to the central role of paying attention with equanimity to one’s body sensations during memory processing.

## Figures and Tables

**Figure 1 ijerph-19-01142-f001:**
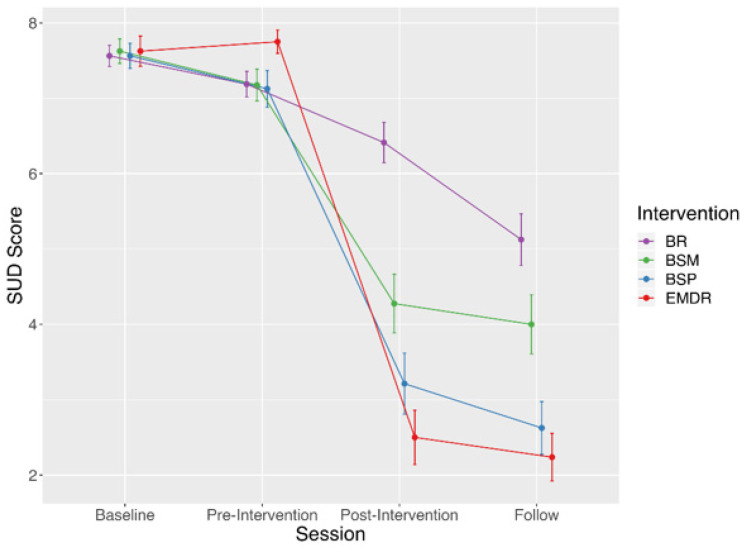
Graph of the interaction effect between Session and Intervention. Notes: t0 = Baseline, t1 = Pre-Intervention, t2 = Post-Intervention, t3 = Follow-up, EMDR = Eye Movement Desensitization and Reprocessing, BSP = Brainspotting, BSM = Body Scan Meditation, BR = Book Reading; Vertical bars denote 0.95 confidence interval.

**Table 1 ijerph-19-01142-t001:** Procedure: Interventions were delivered in an individual setting; each participant received all interventions (one per week, in four consecutive weeks), and each intervention dealt with only one memory of the participant. The order of interventions was counterbalanced among participants (see Methods for details).

Screening Tests	Baseline	Pre- Intervention	Intervention	Post- Intervention	Follow-Up
MCMI-III, ENB2, ROCF					
	Memory telling (×4) SUDS (×4)	Memory telling and SUD	EMDR	Memory telling and SUD	Memory telling (×4) SUDS (×4)
		Memory telling and SUD	BSP	Memory telling and SUD	
		Memory telling and SUD	BSM	Memory telling and SUD	
		Memory telling and SUD	BR	Memory telling and SUD	

**Table 2 ijerph-19-01142-t002:** Williams design: schedule (size = 4 × 4).

		Intervention			
		1	2	3	4
Participants	1, 5, 9, 13, 17, 21, 25, 29, 33, 37	A (EMDR)	B (BSP)	D (BR)	C (BSM)
	2, 6, 10, 14, 18, 22, 26, 30, 34, 38	B (BSP)	C (BSM)	A (EMDR)	D (BR)
	3, 7, 11, 15, 19, 23, 27, 31, 35, 39	C (BSM)	D (BR)	B (BSP)	A (EMDR)
	4, 8, 12, 16, 20, 24, 28, 32, 36, 40	D (BR)	A (EMDR)	C (BSM)	B (BSP)

Notes: Intervention A = Eye Movement Desensitization and Reprocessing (EMDR); Intervention B = Brainspotting (BSP); Intervention C = Body Scan Meditation (BSM); Intervention D = Book Reading (BR).

**Table 3 ijerph-19-01142-t003:** Comparison of the four experimental interventions.

	EMDR	BSP	BSM	BR
Time	41.45 min (SD = 3.41)	40.03 min (SD = 3.59)	40 min	40 min
Eyes	Open	Open	Closed	Open
BLS	Mainly EMs, but also tapping	Only acoustic	None	None
Features of BLS	About 30 s set of fast EMs repeated at intervals	Continuous slow BLS	None	None
Participant’s attention	Focus on stream of consciousness linked to the memory	Open monitoring of the stream of consciousness linked to the memory while focusing primarily on body sensations	Focus on body sensations (regardless of any connection with the distressing memory)	Focus on a distracting task (book reading)
Conversational approach	Limited. Participant reports verbal feedback after each set of BLS. Therapist invites participant to continue relying on the process (e.g., “That’s it”, “Good”, “You’re doing well”, “What did you notice?”, “Go with that”)	Very limited. No need of participant verbal reports. Typically, silent therapist; occasionally, therapist invites participant to bring attention back to the body sensations	None. Participant in silence; therapist verbally guides the meditation	None. Silent reading.
Required relationship features	Cooperative	“Dual attunement”	Guidance, instruction	Not relevant

Notes: EMDR = Eye Movement Desensitization and Reprocessing; BSP = Brainspotting; BSM = Body Scan Meditation; BR = Book Reading; EMs = Eye Movements; BLS = Bilateral Stimulation.

**Table 4 ijerph-19-01142-t004:** Screening test descriptive statistics (*n* = 40).

Screening Test		Min	Max	M	SD
MCMI-III	Schizotypal	0.00	62.00	14.00	21.68
	Borderline	0.00	60.00	7.58	11.49
	Paranoid	0.00	62.00	21.55	22.33
	Thought disorder	0.00	76.00	13.25	16.50
	Major depression	0.00	76.00	10.78	15.72
	Delusional disorder	0.00	65.00	8.45	14.99
ENB2	Long-term verbal memory	14.00	28.00	22.13	3.77
	Phonemic fluency	12.33	27.33	17.80	3.85
ROCF	Equivalent Score	1	4	3.78	0.66

Notes: MCMI-III = Millon Clinical Multiaxial Inventory, ENB2 = Italian Short Neuropsychological Assessment, ROCF = Rey–Osterrieth Complex Figure Test.

**Table 5 ijerph-19-01142-t005:** Subjective Units of Disturbance and Memory Telling Duration raw scores.

*n* = 40	EMDR				BSP				BSM				BR			
	M	SE	CI −95%	CI +95%	M	SE	CI −95%	CI +95%	M	SE	CI −95%	CI +95%	M	SE	CI −95%	CI +95%
SUDs t0	7.63	0.20	7.22	8.03	7.56	0.17	7.23	7.90	7.63	0.16	7.30	7.95	7.56	0.14	7.27	7.85
SUDs t1	7.75	0.16	7.44	8.06	7.13	0.24	6.63	7.62	7.18	0.21	6.75	7.60	7.19	0.17	6.84	7.53
SUDs t2	2.50	0.36	1.77	3.23	3.21	0.41	2.39	4.03	4.28	0.39	3.49	5.06	6.41	0.27	5.87	6.95
SUDs t3	2.24	0.32	1.60	2.88	2.63	0.35	1.92	3.33	4.00	0.39	3.21	4.79	5.13	0.34	4.43	5.82
MTD t0	3.42	0.39	2.63	4.21	3.15	0.35	2.44	3.86	3.64	0.43	2.77	4.51	3.55	0.34	2.86	4.24
MTD t1	4.22	0.62	2.98	5.46	3.55	0.47	2.59	4.51	3.57	0.58	2.40	4.74	3.66	0.40	2.85	4.46
MTD t2	2.41	0.21	1.98	2.84	2.24	0.20	1.84	2.64	2.92	0.44	2.04	3.80	3.20	0.36	2.47	3.92
MTD t3	2.49	0.26	1.96	3.01	2.18	0.23	1.72	2.64	2.25	0.29	1.66	2.83	2.48	0.21	2.05	2.90

Notes: t0 = Baseline, t1 = Pre-Intervention, t2 = Post-Intervention, t3 = Follow-up, SUDs = Subjective Units of Disturbance scores (values from 0 to 10), MTD = Memory Telling Duration (expressed in min); EMDR = Eye Movement Desensitization and Reprocessing, BSP = Brainspotting, BSM = Body Scan Meditation, BR = Book Reading, SE = Standard Error of the means, CI = Confidence Interval.

## Data Availability

The data that support the findings of this study are available on request from the corresponding author, Fabio D’Antoni.

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
