# Peer review of "Psychotherapeutic Techniques for Distressing Memories: A Comparative Study between EMDR, Brainspotting, and Body Scan Meditation"

_ijerph, 2022, doi:10.3390/ijerph19031142_

Round 1

Reviewer 1 Report

The study aimed at exploring the effects of a single-session interventions of Eye Movement Desensitization and Reprocessing (EMDR), Brainspotting (BSP) and Body-Scan Meditation (BSM), compared with an active control condition, on negative life event processing reported by a non-clinical sample of adult participants (i.e., psychologists and medical doctors attending a specialization in psychotherapy). Intervention comparative effectiveness was tested in a within-subject paradigm using Subjective Units of Disturbance (SUD) and Memory Telling Duration  (MTD) measures in 4 sessions: Baseline, Pre- and Post-Intervention, Follow-up (about two months after the intervention). Beyond some advantages offered by specific techniques over others, overall, results showed beneficial effects of a single session of EMDR, BSP or BSM in the processing of healthy adults’ distressing memories.

The study is well-designed (including an adequate follow-up), carefully controlled (within-subject design) and compares techniques that have never been compared before, thus it allows the Authors to draw some conclusions in relation to intervention comparative effectiveness in reducing the discomfort of negative memories. Although it can be considered a kind of pilot study (especially given the peculiarity of the sample tested), I think that it can offer a valid contribution to the literature, which needs more than ever studies that directly compare various interventions having similar healing goals.

MAJOR POINTS

I only have some concerns about the sample selected: psychologists and medical doctors attending a specialization in psychotherapy, that is to say participants that, more than others, can have specific beliefs and expectations with respect to the effectiveness of the various techniques used (especially when these can be- or could have been- training topics of their specialization course). Indeed, this could have increased their positive anticipations on the experimental techniques, compared with the control condition (which is easily detectable for a professional of the field). In turn, this could have  led to a kind of amplified placebo effect with a possible magnification of results in favour of the experimental techniques. In this regard, collecting participants’ beliefs on every technique effectiveness before their enrolment could have helped in discarding the participants too “favourable”, or, at least, to control the outcomes for baseline anticipations. The Authors have already discussed this limit in the discussion section, but I suggest to give more emphasis to the issue (hinting at the possible undesired effects of this sample selection on results), and to further highlight the need for study replications in “non-professionals” samples.

I suggest to use a more cautious approach to the topic avoiding the use of the label “psychotherapeutic” when referring to these techniques, since there is not a general agreement on this. I would suggest to mention them as techniques that can be integrated in a psychotherapeutic intervention, saving the term “psychotherapeutic” only for interventions, generally more complex and structured, requiring a specialization in psychotherapy to be delivered

“Results” section: In my opinion, the part on “SUD for memories related to potentially traumatic major (“T”) and relational (“t”) events” is not essential to the study and does not enrich it (at least not in the present form). Specifically, my concern is related to the fact that the classification of participants’ experiences as traumatic or non-traumatic has been made by judges on the basis of a presumed “objective value of traumaticity” of a given event, which can be quite different from the personal value that each participant could have given to the same event. It is true that some classification system use “objective” criteria to define what can be termed “trauma” (e.g., see the various versions of DSM), however, I wonder why the Authors did not consider classifying the participants' experiences as less or more traumatic based on “subjective” parameters, i.e., parameters coming from participants (like for instance, a “disturbing/upsetting value” given to each event or to each memory by the participant who experiences that event). Thus, also in the light of other limits described by the Authors for this part (“results should be taken with caution because of the non-homogeneous representation of “T” and “t” within the four conditions”), I would suggest either remove it or highlight this further limitation.

“Discussion” and “Conclusions” section: Results shows that EMDR and BSP were comparable in terms of effectiveness in distressing memories re-processing (effectiveness which was somehow superior to that of BSM). The Author stated that “these results were primary attributed to the central role of paying attention with equanimity to one’s body sensations during memory processing”. Someone can agree with this opinion, however, EMDR and BSP share another important feature which has been neglected: both the techniques (unlike the other) entail a “stream of consciousness linked to the memory”. Given that this “stream of consciousness” implies a kind of imaginative exposure to the distressing memory (exposure that some believe to be responsible of EMDR therapeutic effects), I think it is important to hypothesize that imaginative exposure can be another possible factor responsible of the superiority of these two techniques (remarkably, given that the control condition of reading about traumas may have activated the memory of one’s own distressing event, thus producing a limited imaginative exposure to it, the factor could explain also the SUD score reductions in the control condition).

MINOR POINTS:

“Materials and methods” section: the first time in which MTD is introduced, it should be mentioned how the length of time spent in recounting the memory is interpreted (in term of  its functional vs. dysfunctional memory processing), although this is inferable in the following sections.

“Materials and methods” section: it is not clear to me whether the SUD measure is completed the same number of times in all of the conditions. If this is not the case and it was completed more times for the EMDR technique (or for others) it should be mentioned, given that the repetition of the evaluation act could affect the variable evaluated

In table 2, there is some round bracket missing

Please, give more details on the meaning of “Dual attunement” for BSP

Why the Authors chose a book about traumatic stress for their control condition rather than a book with a neutral content? I mean, could this choice have led the participants to implicitly equate their stressful- but not traumatic- experience to a more serious trauma (the topic of the book), thus contributing to evaluate its relative memory as still disturbing at post-intervention?

Author Response

The study aimed at exploring the effects of a single-session interventions of Eye Movement Desensitization and Reprocessing (EMDR), Brainspotting (BSP) and Body-Scan Meditation (BSM), compared with an active control condition, on negative life event processing reported by a non-clinical sample of adult participants (i.e., psychologists and medical doctors attending a specialization in psychotherapy). Intervention comparative effectiveness was tested in a within-subject paradigm using Subjective Units of Disturbance (SUD) and Memory Telling Duration (MTD) measures in 4 sessions: Baseline, Pre- and Post-Intervention, Follow-up (about two months after the intervention). Beyond some advantages offered by specific techniques over others, overall, results showed beneficial effects of a single session of EMDR, BSP or BSM in the processing of healthy adults’ distressing memories.

The study is well-designed (including an adequate follow-up), carefully controlled (within-subject design) and compares techniques that have never been compared before, thus it allows the Authors to draw some conclusions in relation to intervention comparative effectiveness in reducing the discomfort of negative memories. Although it can be considered a kind of pilot study (especially given the peculiarity of the sample tested), I think that it can offer a valid contribution to the literature, which needs more than ever studies that directly compare various interventions having similar healing goals.

 MAJOR POINTS

  1. I only have some concerns about the sample selected: psychologists and medical doctors attending a specialization in psychotherapy, that is to say participants that, more than others, can have specific beliefs and expectations with respect to the effectiveness of the various techniques used (especially when these can be- or could have been- training topics of their specialization course). Indeed, this could have increased their positive anticipations on the experimental techniques, compared with the control condition (which is easily detectable for a professional of the field). In turn, this could have led to a kind of amplified placebo effect with a possible magnification of results in favour of the experimental techniques. In this regard, collecting participants’ beliefs on every technique effectiveness before their enrolment could have helped in discarding the participants too “favourable”, or, at least, to control the outcomes for baseline anticipations. The Authors have already discussed this limit in the discussion section, but I suggest giving more emphasis to the issue (hinting at the possible undesired effects of this sample selection on results), and to further highlight the need for study replications in “non-professionals” samples.

Thanks to Reviewer 1 for providing us these helpful and detailed suggestions. Based on what s/he suggested we reviewed the discussion section accordingly. In particular, we gave more emphasis to the possibility of undesired effects of our sample selection on results as follows: “Indeed, participants were psychologists and medical doctors attending a specialization in psychotherapy and their beliefs and expectations with respect to the effectiveness of the various techniques used could have affected results (e.g., amplifying the effectiveness of experimental techniques compared with the control condition). However, participants’ possible positive anticipations on the experimental techniques might explain only partially the outcomes of the present study since EMDR and BSP were found to be more effective not only than BR but also BSM. Nonetheless, to overcome this limitation concerning the selected sample, it would be useful that future studies attempt to extend the present findings to larger groups of “non-professionals” participants who, with caution, could be recruited outside psychotherapeutic trainings” (p. 13).

  1. I suggest to use a more cautious approach to the topic avoiding the use of the label “psychotherapeutic” when referring to these techniques, since there is not a general agreement on this. I would suggest to mention them as techniques that can be integrated in a psychotherapeutic intervention, saving the term “psychotherapeutic” only for interventions, generally more complex and structured, requiring a specialization in psychotherapy to be delivered

We have replaced the word "psychotherapeutic" with "therapeutic" as suggested or referred to the three techniques used in our study as “techniques that can be integrated in a psychotherapeutic intervention” (e.g., p. 2).

  1. “Results” section: In my opinion, the part on “SUD for memories related to potentially traumatic major (“T”) and relational (“t”) events” is not essential to the study and does not enrich it (at least not in the present form). Specifically, my concern is related to the fact that the classification of participants’ experiences as traumatic or non-traumatic has been made by judges on the basis of a presumed “objective value of traumaticity” of a given event, which can be quite different from the personal value that each participant could have given to the same event. It is true that some classification system use “objective” criteria to define what can be termed “trauma” (e.g., see the various versions of DSM), however, I wonder why the Authors did not consider classifying the participants' experiences as less or more traumatic based on “subjective” parameters, i.e., parameters coming from participants (like for instance, a “disturbing/upsetting value” given to each event or to each memory by the participant who experiences that event). Thus, also in the light of other limits described by the Authors for this part (“results should be taken with caution because of the non-homogeneous representation of “T” and “t” within the four conditions”), I would suggest either remove it or highlight this further limitation.

We agree with the suggestion to remove the section concerning the distinction of traumatic memories in "T" and "t" because it does not significantly enrich the present study. Thus, we removed this brief section from the revised manuscript. 

  1. “Discussion” and “Conclusions” section: Results shows that EMDR and BSP were comparable in terms of effectiveness in distressing memories re-processing (effectiveness which was somehow superior to that of BSM). The Author stated that “these results were primary attributed to the central role of paying attention with equanimity to one’s body sensations during memory processing”. Someone can agree with this opinion, however, EMDR and BSP share another important feature which has been neglected: both the techniques (unlike the other) entail a “stream of consciousness linked to the memory”. Given that this “stream of consciousness” implies a kind of imaginative exposure to the distressing memory (exposure that some believe to be responsible of EMDR therapeutic effects), I think it is important to hypothesize that imaginative exposure can be another possible factor responsible of the superiority of these two techniques (remarkably, given that the control condition of reading about traumas may have activated the memory of one’s own distressing event, thus producing a limited imaginative exposure to it, the factor could explain also the SUD score reductions in the control condition).

We thank the reviewer for this cogent observation on which we agree. We added the hypothesis on the possible role of imaginative exposure in contributing to explain the effectiveness of EMDR and BSP (compared to BSM) in distressing memories re-processing as follows: “Another possible explanation of the effectiveness of, in particular, EMDR and BSP in reducing SUD scores connected to distressing memories may be connected to imaginative exposure. Indeed, although not to the same extent as with other techniques such as exposure therapy, these two techniques imply a kind of imaginative exposure to the distressing memories, and this can contribute to their adaptive re-processing53. Remarkably, such a possibility could also explain, at least in part, the reduction of SUD scores in the control BR condition. Indeed, reading a book about trauma may have triggered the memory of one’s own distressing event, thus producing a limited imaginative exposure to it” (p. 12).

We also added a reference as follows: “53. Rogers, S.; Silver, S. M. Is EMDR an Exposure Therapy? A Review of Trauma Protocols. J. Clin. Psychol. 2002, 58 (1), 43–59. https://doi.org/10.1002/jclp.1128.” (p. 16)

MINOR POINTS:

  1. “Materials and methods” section: the first time in which MTD is introduced, it should be mentioned how the length of time spent in recounting the memory is interpreted (in term of its functional vs. dysfunctional memory processing), although this is inferable in the following sections.

We introduced in the “Materials and methods” section how the length of time spent in recounting the memory is interpreted as follows: “This measure was hypothesized to be connected to the SUDS assuming that, as the memory processing took place, it was possible to find both a decrease of SUD scores and a reduced length of time spent to recounting the memory.” (p. 5)

Note: in the discussion section we proposed a possible account of why memory processing may be reflected in fewer words used to describe it.

  1. “Materials and methods” section: it is not clear to me whether the SUD measure is completed the same number of times in all of the conditions. If this is not the case and it was completed more times for the EMDR technique (or for others) it should be mentioned, given that the repetition of the evaluation act could affect the variable evaluated

We clarified better the use of the SUDS as follows: “The SUDS were completed the same number of times in all of the conditions. During memory processing with EMDR and BSP we used generally the term "activation" instead of “distress/disturbance” (see below for more details on EMDR and BSP interventions) (p. 5).

  1. In table 2, there is some round bracket missing

We inserted the missing round brackets in Table 2. 

  1. Please, give more details on the meaning of “Dual attunement” for BSP

We added more details on the meaning of “Dual attunement” for BSP as follows: “In Brainspotting therapy, the expression "dual attunement" refers to a process supposed to be both relational and neurological, through which the therapist continuously tries to remain connected to the therapeutic relationship as well as to the client brain-body response in therapy. According to this approach the attuned, empathic, witnessing presence of the therapist promotes adaptive changes in the client23” (p. 7) 

  1. Why the Authors chose a book about traumatic stress for their control condition rather than a book with a neutral content? I mean, could this choice have led the participants to implicitly equate their stressful- but not traumatic- experience to a more serious trauma (the topic of the book), thus contributing to evaluate its relative memory as still disturbing at post-intervention?

We thank the reviewer for this interesting observation. We highlighted the possible limitation in using a non-neutral control condition as follows: “Another limitation of our study may be related to the choice of the book used in the control condition. Indeed, reading a book on trauma may have led participants to implicitly equate their distressing, but not traumatic, memories to more serious traumas, thus contributing to evaluate their memories as still disturbing at the post-intervention phase. Therefore, it would be useful to carry out further studies with other neutral control conditions.” (p. 13).

Reviewer 2 Report

See attached file!

Author Response

Review IJERPH manuscript 1544604

This is a very interesting study on the effect of intervention on distressing memories in a non-clinical sample with EMDR, BSP, BSM. The methods are, on the whole, well described.

The manuscript is basically well written and there is a sound research design. However, some amendments are needed before it could be accepted for publication.

  1. Abstract: the comparison between the interventions are less clearly described compared to the result paragraph. Try to clarify!

Thanks to Reviewer 2 for providing us with many stimulating suggestions to improve our manuscript. In the abstract, we followed his/her recommendation trying to clarify the comparison between the interventions as follows: “Participants (N = 40 Psychologists/MDs) reported four distressing memories, each of which treated with a single intervention. EMDR, BSP and BSM were compared with each other and with a Book-Reading (BR) control condition using as dependent measures SUD (Subjective Units of Disturbance) and Memory Telling Duration (MTD) on a 4-point timeline: Baseline, Pre-Intervention, Post-Intervention, Follow-up.” (p. 1)

  1. Introduction: The section is well written, and the methods well described. However, the control intervention with BR (book reading) is not introduced. There is no reference in the text, no explanation, why it was chosen, and it is not mentioned in the aims.

We explained why “Book Reading” intervention was chosen as control condition as follows: “To evaluate this hypothesis, we also introduced an active control condition, in which participants were engaged in reading a book about trauma, that is in an activity not designed to target the core therapeutic features of interest.” (p. 3)

  1. Furthermore, why did the authors choose a non-clinical sample? Any specific purpose? I can see possible advantages, but what was the actual intention?

We specified the reasons why we chose a non-clinical sample as follows: “A non-clinical sample was chosen because we intended to reduce any potential negative effect of eventual severe psychopathology, neuropsychological signs, or psychopharmacological treatment, on both the therapeutic relationship and the capacity of participants to self-focusing on their inner experience and recollecting memories; moreover, we also intended to limit the possible adverse effects potentially originating from the (brief) treatment of participants' distressing memories.” (p. 3).

  1. Material & method: the inclusion criteria are well described, but how many persons were approached. How many aggreed? Consecutive cases?

We added more details on participants as follows: “We approached 48 persons, 42 of whom initially agreed to participate in the research project. Before the actual start of the project a participant withdrew from the research while the last participant agreed to be not included in the study due to the completion of the Williams experimental design reached at the fortieth participant.” (p. 4).

  1. I agree that the professional experience of the participants may be an advantage (p 4 first paragraph). The measures and procedures are on the whole well described. I the result section, events are divided into traumatic (T) and relational (t)events. This dicotomy should be introduced i M&M.

According to suggestions of Reviewer 1, we removed the brief section concerning the distinction of traumatic memories in "T" and "t". We agree that it did not significantly enrich the present study.

  1. Results: on the whole well written. The figure gives a good illustration. Discussion: well written and major limitations are acknowledged. The impact of the experience of the therapist is discussed. Make sure this topic is also acknowledged in M&M, so we know something about this. Are the therapist’s experience compatible between the methods? Could bias possibly have been introduced?

We rephrased the original text as follows: “The therapist was thus trained at an advanced level in each of the three techniques used in the study, which, moreover, are commonly used by him in his ordinary clinical activity.” (p. 8).

  1. Conclusion: any suggestion for further research on a clinical sample

We added some suggestions for further research on clinical samples as follows: “Thus, this study underlines the importance of conducting further research comparing the effects of different therapeutic interventions on the processing of distressing/traumatic memories in clinical populations. For example, it would be interesting to study which specific features of therapeutic interventions could be most useful for each specific personality disorder. From this point of view some questions may arise: could EMDR, as a structured protocol, potentially be more useful than BSP in borderline personality patterns? Or might it be more useful to use the more “open”, less structured, approach of BSP with avoidant clients? Or are there times depending on the development of the therapeutic relationship for which it would be desirable to use first one kind of intervention and then another one? Future research may help answer questions like these.” (p. 13).
